# Food Insecure College Students and Objective Measurements of Their Unused Meal Plans

**DOI:** 10.3390/nu11040904

**Published:** 2019-04-23

**Authors:** Irene van Woerden, Daniel Hruschka, Sonia Vega-Lόpez, David R. Schaefer, Marc Adams, Meg Bruening

**Affiliations:** 1College of Nursing, Idaho State University, Pocatello, ID 83209, USA; 2School of Human Evolution and Social Change, Arizona State University, Tempe, AZ 85287, USA; dhruschka@asu.edu; 3College of Health Soultions, Arizona State University, Phoenix, AZ 85004, USA; sonia.vega.lopez@asu.edu (S.V.-L.); marc.adams@asu.edu (M.A.); meg.bruening@asu.edu (M.B.); 4Department of Sociology, University of California, Irvine, CA 92697, USA; drschaef@uci.edu

**Keywords:** food insecurity, college, university, students, freshmen, meal plans, dining halls

## Abstract

Some researchers have proposed the prevalence of food insecurity among college students is high due to students’ meal plans providing insufficient meals. The association between college students’ food security status and their meal plans have not yet been examined. In this study, United States (US) first year college students (*N* = 534) self-reported their food security status in the Fall 2015 and/or Spring 2016 semester(s). Objective measures of students’ meal plans were obtained from the university. Logistic generalized estimating equations (GEEs) were used to determine if students’ meal plan, and meal plan use, predicted food insecurity. Linear GEEs were used to examine several potential reasons for lower meal plan use. We found that students did not use all of their available meals. Compared to students on the most expensive (unlimited) meal plan, students on the cheapest (8 meals/week) meal plan were the most likely to report food insecurity (OR = 2.2, 95% CI = 1.2, 4.1). However, in the Fall semester, 26% of students on unlimited meal plans also reported food insecurity. For students on the 180 meals/semester meal plan, food insecurity was associated with using fewer meals (OR = 0.9, 95% CI = 0.8, 1.0). Students who worked tended to use their meal plan less (β = −1.3, 95% CI = −2.3, −0.3). Students are reporting food insecurity while having meals left in their meal plan.

## 1. Introduction

Food security is defined as access at all times to enough food for an active, healthy life [1], whereas food insecurity indicates inconsistent access to healthy food. Systematic reviews estimate the prevalence of food insecurity among United States (US) college students at 33% [2] and 51% [3], much higher than for US households (11.8%) [4]. The high prevalence of food insecurity among college students is of concern because food insecurity is associated with lower dietary quality [5,6,7,8], worse health outcomes [9,10,11], and lower academic performance [12,13,14,15,16]. For instance, college students with food insecurity are more likely to report higher stress [13,17], higher depression [13,17,18,19], less sleep [19], and a lower grade point average (GPA) [12,13,14,15,16] relative to their food secure peers. 

Students who report food insecurity are also more likely to be financially independent from their family [15,19,20], to have debt [15], and to receive financial aid [17,19,20] than their food secure counterparts. It is unclear whether food insecure students tend to purchase less expensive meal plans than food secure students, and if food insecure students use their meal plan to a lesser (or greater) extent than their food secure counterparts. The extent to which food secure and insecure students use their meal plans is of interest as student meal plans are typically prepaid at the start of the semester; students on the same meal plan have the same number of meals available to them. 

In the US, many first-year college students are required to live on campus and to purchase a meal plan. The meal plans that universities offer typically consist of either a fixed number of dining hall meals per week or semester, or unlimited dining hall meals. Some meal plans also have cash-equivalence for food/beverage purchases at other, non-dining hall outlets on campus. Students are expected to use their meal plan for the majority of their meals; students often do not have access to a full kitchen as they are not expected to cook for themselves. In general, if a student does not use all of the meals provided by their meal plan, the remaining meals are forfeited. There is currently limited information regarding the number of meals that students forfeit.

Some studies suggest that students experience food insecurity because their meal plans are insufficient, while other studies report no association between food insecurity and college students’ meal plans. Studies describe a student on a 14 meals per week plan going hungry for a few days after using all of their meals early in the week [21], and a student who ran out of meals on their meal plan [22]. Another study found that 43% of students who had a meal plan also reported food insecurity [23]. Researchers and university students and staff have suggested that changing the current structure of meal plans could help to decrease the rate of food insecurity among students. For instance, Gallegos suggests providing students a way to pay for their meals through their taxes once they are employed [12] and university staff have suggested extending the dining hall hours [24] (dining halls typically open in the morning and close in the evening). University students and staff have also suggested more basic, affordable, meal plan options may be beneficial [24]. One of the studies that reported no association between food insecurity and meals plans found that first-year college students on a basic meal plan (8 meals/week) were no more likely to report food insecurity over the prior three months than first-year students on a meal plan that provided more meals per week [18]. Three other studies examined if having a meal plan (compared to not having a meal plan) was protective of food insecurity and found null results; two of the studies included students of all year levels [14,25], while the third study excluded first-year and graduate students [20]. 

There are also indications that many students do not use all of their meal plan meals. Newspaper articles have suggested that food service providers price the meal plans with an expectation of 60-70% of the meals provided by each meal plan being used [26], and that roughly the same proportion of meal plans are actually used [27]. It is clear that some students have unused meals left on their meal plan: The program Swipe Out Hunger redistributes students’ unused meals to others and reports providing 1.5 million dining hall meals since it was established in 2010 [28]. If students experiencing food insecurity also report unused meals, then understanding why this occurs could help to explain why so many students report food insecurity.

The extent to which students use their meal plans may change by students’ mental health, time commitments, and alternative food supplies; these same variables may also be associated with students’ food security status. For instance, food insecurity is associated with anxiety and depression [17], which in turn can interfere with daily life [29,30]. Students experiencing anxiety and depression may withdraw from the dining hall environment and as such have lower meal plan use. Prior studies have found that food insecure students are more likely to work a job than food secure students [14,16,19,25]. Working a job may make it difficult for a student to use their meal plan as they may be unable to access the dining halls during opening hours. Inconvenient hours have previously been reported as a reason why those with food insecurity do not use food pantries [31,32], and students have also reported scheduling conflicts with classes which prevent them from using the dining halls [21]. It has been reported that students have a substantial number of food items in their residence hall room [33]. Students may find it more convenient to eat the food in their room rather than to eat at the campus food outlets. 

This study examines college students’ food security status and meal plans. Food insecurity by itself is of clinical concern [5,6,7,8,9,10,11,13,17,18,19]. As meal plan use is dietary-related behavior, this study sets the stage for future research to assess differential dietary intake by differential meal plan use. First, the association between students’ food security status and type of meal plan (i.e., weekly, semester, unlimited) purchased was examined. Then, objective data were used to determine if students who purchase meal plans that provide fewer meals and students who use their meal plan less are more likely to report food insecurity. Finally, several potential reasons that could result in lower meal plan use (anxiety, depressed mood, working a job, perceived meal plan inadequacy, and alternative food supplies) were examined. It was hypothesized that food insecure students and students with anxiety or depressed mood, who worked a job, who perceived their meal plan provided inadequate food, and with alternative food supplies would have lower meal plan use. As students may be off-campus during the weekend, and as such unable to use their meal plan, or have no need to use their meal plan, the extent to which each student used their meal plan during the weekdays and during the full week was differentiated. This information can help to shape future interventions aimed at minimizing food insecurity on college campuses. 

## 2. Materials and Methods

This is a secondary analysis from the larger, longitudinal social impact of physical activity and nutrition in college (SPARC) study conducted at Arizona State University (ASU) [34]. The SPARC survey included measures of the prior month’s food security status, mental health, employment status, alternative food supplies, and meal plan adequacy. Students participating in the SPARC study also consented to the researchers accessing their university-collected data. The university provided information on the type of meal plan each student purchased, and their dining hall use. A total of 1435 students participated in the SPARC study during the 2015–2016 academic year (Fall 2015 semester: August–December, Spring 2016 semester: January–May). A total of 534 SPARC participants were first-year students living on one of the two targeted campuses (and not in the honors college) with usable information at the end of the Fall 2015 (*n* = 470), or Spring 2016 (*n* = 324) semesters; 260 students had usable information for both the Fall and Spring semesters. Participants were recruited primarily from floor meetings at residence halls at the start of the academic year. All participants provided written informed consent. Study protocols were approved by the Arizona State University Institutional Review Board.

### 2.1. Measurements

Food security status. The United States Department of Agriculture (USDA) six-item food security short form [35] was used to determine students’ past month’s food security status at the end of each semester. Participants were asked to indicate how often the following statements were true in the last one month: “The food that I bought just didn’t last, and I didn’t have money to get more”, “I couldn’t afford to eat balanced meals”. The response options were “often true”, “sometimes true”, and “never true”. Students were then asked “In the past 1 month, did you ever cut the size of your meals or skip meals because there wasn’t enough money for food?” The response options were “No”, “Yes, only 1 or 2 weeks”, “Yes, some weeks but not every week”, and “Yes, almost every week”. Finally, students were asked to indicate if the following statements were true for them in the last 1 month: “Did you ever eat less than you felt you should because there wasn’t enough money for food?”, “Were you ever hungry but didn’t eat because there wasn’t enough money for food?” The response options were “Yes” and “No”. Any student who responded affirmatively to two or more questions was recorded as being food insecure.

Meal plan and dining hall use. The university provided the meal plan used and date/time stamp for when each participant used their meal plan to enter a dining hall or to purchase a meal exchange. Other than a few days with zero recorded meal plan use, students’ meal plan use was available for the entire semester. Zero recorded meal plan use was observed for five days during the Fall 2015 semester (one of which was Thanksgiving Day, confirmation that the system down for three of the other days was obtained), and for four consecutive days during the Spring 2016 semester (one of which was Martin Luther King Day). It was unclear if the absence of meals recorded on the four days in Spring was due to a system failure during those days, subsequent data corruption, or another reason. 

At this university, the four meal plans available to students are 8 dining hall meals/week, 14 dining hall meals/week, 180 dining hall meals/semester, and unlimited dining hall meals [36]. Each meal plan provided students with one “meal exchange” per day which allowed students to purchase one on-the-go meal (such as a sandwich, chips, and a drink) at a few of the on-campus food outlets [37]. While each of these meal plans also includes some cash-equivalence that can be used at on-campus food outlets, the dataset used for this analysis did not include students’ cash-equivalence use. Students can officially only change their meal plan during the first week of a semester [37]; however, some students appear to have changed their meal plan after this deadline—the meal plan some students used to access the dining hall changed after the first week. The reasons students changed their meal plan are unknown. Participants who changed their meal plan were classified as using the meal plan they used after the third week of the semester. Any student whose meal plan changed after the third week of the semester was excluded from the analyses for that semester (*n* = 9). There were three students with no meal plan use in the 28 days prior to survey completion but who used their meal plan after they completed the survey and prior to the semester ending. These three students were included in the analysis and recorded as having zero use during the 28 days prior to survey completion. 

Mental health. Participants’ depressed mood was examined using the 2013 American College Health Association survey [38]. At the end of each semester, participants reported how often in the past one month they had felt “things were hopeless”, “overwhelmed by all they had to do”, “very lonely”, “very sad”, “so depressed that it was difficult to function”, and “overwhelming anxiety”. The response options were “never”, “rarely”, “sometimes”, and “often” and coded to a 1 (never)–4 (often) scale. Participants whose average response to these 6 items was above 2.0 were classified as having high levels of depressed mood. Participants’ anxiety was determined by asking “In the past 12 months, have you been told by a doctor or health care professional that you have anxiety?” [39] The response options were “No”, “Yes, diagnosed and treated”, and “Yes, diagnosed”. Participants who reported a diagnosis of anxiety within the past 12 months were classified as having anxiety.

Working a job. At the end of each semester, participants were asked “On average, how many hours a week do you work for pay?”, a question also created specifically for the SPARC study. The response option was free flow text. Based on the distribution of responses, the responses were categorized as working for pay (any number of hours) and not working.

Meal plan adequacy. At the end of each semester, participants were asked “Do you feel like your meal plan provides you with adequate food each week?”, a question created specifically for the SPARC study. The response options were “strongly disagree”, “disagree”, “agree”, and “strongly agree” and were further categorized as “strongly disagree/disagree” and “strongly agree/agree”.

Alternative food supplies. Three questions were asked to determine participants’ alternative food supplies. These items were created specifically for the SPARC study. At the end of each semester, participants were asked, “How often do your parents/guardians typically purchase or send food for you?” The response options were “never”, “once per semester”, “2–3 times per semester”, “monthly”, and “weekly” and were categorized as “less than twice a semester” and “twice a semester or more”. Participants were also asked “When thinking about your meal plan, how often do your friends share their meal plan with you?” and “When thinking about your meal plan, how often does your roommate/suitemate(s) share their meal plan with you?” The response options for both questions were “never”, “rarely”, “sometimes”, and “often” and were further classified as “never/rarely” and “sometimes/often”. 

Sociodemographics, The University provided participants’ sex, race/ethnicity, date of birth, and family economic status. Students self-reported if they were a first-generation college student and which campus they resided on. The University administration recorded students’ sex as male and female and students’ race/ethnicity as “American Indian/Alaska Native”, “Asian”, “Black/African American”, “Hispanic/Latino”, “International”, “Native Hawaiian/Pacific Islander”, “Two or More Races”, “Unspecified”, and “White”. Due to low counts for American Indian/Alaska Native, Asian, International, Native Hawaiian/Pacific Islander, Two or More Races, and Unspecified groups, these classifications were combined to “Other” race/ethnicity. The race/ethnicities used in the current analyses are “White”, “Hispanic/Latino”, “Black/African American”, and “Other”. Participants’ age was determined based on their date of birth. Students’ family economic status was determined by whether the student was a Pell Grant recipient or not (a Pell Grant is federal aid provided to low-income students [40]). All students self-reported the highest degree or level of education that their parents/guardians completed. The response options were “Some high school (no degree)”, “High school diploma (or equivalent)”, “Some college (no degree)”, “Associate’s degree/Trade/Technical/Vocational training”, “Bachelor’s degree”, “Graduate or professional degree”, and “Not applicable”. If a student reported that neither parent/guardian had a bachelor’s, graduate, or professional degree, the student was classified as a first-generation college student. Eight students reported being involved with an ASU varsity sports team in at least one semester. Due to the risk of identifying students, and the small sample size, no analyses on varsity sports teams was conducted

### 2.2. Statistical Analyses

Bivariate associations between the meal plan purchased and the key variables were examined for each semester using chi-square and ANOVA tests as appropriate. The Fall 2015 and Spring 2016 semesters were examined separately as students’ decisions regarding their meal plans were anticipated to change as they became more familiar with the university and their dining options. The percentage of available meals used in the 28 days prior to survey completion, and across the semester, was determined separately for males and females on each meal plan: The 180 meals/semester meal plan equates to approximately 11 meals per week with constant use; a conservative estimate of 16 weeks for the Fall 2015 semester and 17 weeks for the Spring 2016 semester was used. 

A logistic generalized estimated equation (GEE) was used to examine if students’ food security status at the end of the Fall and/or Spring semesters was predicted by the meal plan a student purchased, after controlling for food security status at the start of the corresponding semester (i.e., food security at the end of Fall predicted by Fall meal plan and food security status at the start of Fall). To capture changes in meal plan use at the end of the semester, the week the survey was completed was included in the model. A clustering effect by student was included to account for the same students responding in the Fall and Spring semesters. Controls for sex, race/ethnicity, Pell Grant status, first-generation status, campus, and semester were also included.

For each meal plan type, logistic GEEs were used to examine if food security status at the end of the Fall and/or Spring semester was predicted by the number of times a student used their meal plan in the 28 days prior to the students’ survey completion. As the number of meals the student used for themselves, and not provided to guests, was of interest, meal plan use was determined as the number of meals the student used excluding guest passes. To aid in the interpretation of meal plan use, meal plan use in the prior 28 days (four weeks) was divided by four to obtain a weekly average. Controls for food security status at the start of the corresponding semester, sex, race/ethnicity, Pell Grant status, first-generation status, campus, semester, week survey was completed, and repeated measures by student were included in the models. Logistic GEEs stratified by sex were then run to determine if, when examining males and females separately, students’ food security status at the end of the Fall and/or Spring semester was predicted by the number of times a student used their meal plan in the 28 days prior to the students’ survey completion. Given the smaller sample size for these stratified models, only controls for food security status at the start of the corresponding semester, campus, semester, and a clustering effect were included in the model.

A linear GEE was used to determine if the number of meals a student used on average per week in the 28 days prior to completing the survey at the end of the Fall and/or Spring semester was predicted by students’ self-reported mental health (depressed mood, anxiety), employment status, alternative food supplies (from parents, roommates, and friends), and perceived meal plan adequacy, as measured at the end of the corresponding semester. Controls for sex, race/ethnicity, Pell Grant status, first-generation status, campus, semester, semester, and repeated measures by student, were included in the models. Linear GEEs, stratified by sex, were run to determine if the associations between number of meals used and students’ self-reported mental health, employment status, alternative food supplies, and perceived meal plan adequacy, were consistent when examining males and females separately. Given the smaller sample size for the stratified models, only controls for campus, semester, and a clustering effect were included in the model.

## 3. Results

The prevalence of food insecurity was 36% at the end of the Fall 2015 semester and 35% at the end of the Spring 2016 semester. No bivariate association with food insecurity and meal plan was found in the Fall semester; the prevalence of food insecurity at the end of the Fall semester ranged from 41% of students on the cheapest meal plan to 26% of students on the unlimited meal plan (*p* = 0.135; Table 1). However, students’ food security at the end of the Spring semester was associated with students’ meal plan; the prevalence of food insecurity ranged from 47% of students on the cheapest meal plan to 11% of students on the most expensive meal plan (*p* = 0.001). 

At the bivariate level, the more expensive meal plans were consistently more likely to be purchased by males (*p* < 0.001), and less likely to be purchased by Pell Grant recipients (*p* < 0.001 Fall, *p* = 0.019 Spring), first-generation college students (*p* < 0.001 Fall, *p* = 0.001 Spring), and students whose parents sent them food at least twice a semester (*p* < 0.001; Table 1). In the Fall semester, students on the cheapest meal plan were the most likely to be working (33% vs 20% to 25%, *p* = 0.041) and the least likely to report that their meal plan provided adequate food each week (67% vs 81% to 87%, *p* = 0.002); however, these associations were not seen in the Spring semester (45% vs 34% to 45%, *p* = 0.645 for working a job; 74% vs 76% to 83%, *p* = 0.679 for meal plan adequacy).

Students did not fully use their meal plans. Students on more expensive meal plans used their meal plan more frequently (*p* < 0.001; Table 2). Female students on the 8 meals/week meal plan used 71% and 64% of their available meals in the 28 days prior to survey completion in the Fall and Spring semesters, respectively; for female students on the 14 meals/week meal plan, these percentages drop to 60% (Fall) and 53% (Spring). In the 28 days prior to survey completion, male students on the 8 meals/week meal plan used 78% and 75% of their available meals in the Fall and Spring semesters, respectively; male students on the 14 meals/week meal plan used 74% and 67% of their available meals in the Fall and Spring semesters, respectively. On average, female (male) students on the 180 meals per semester meal plan used 62% (68%) of their meals in the Fall semester and 70% (74%) of their meals in the Spring semester.

The adjusted odds ratio of food insecurity was significantly greater for students on the 8 meals/week (OR = 2.2, 95% CI = 1.2, 4.1) plan than for students on the unlimited meal plan (Table 3). 

When examining females and males together, students on the 180 meals/semester meal plan who used fewer meals per week were more likely to report food insecurity at the end of the semester; the odds of food insecurity decreased as students used more meals when weekends were included (OR = 0.9, 95% CI = 0.8, 1.0) and excluded (OR = 0.8, 95% CI = 0.7, 0.9; Table 4). However, when stratifying by gender, the association between meal plan use and food insecurity was only seen among males on the 180 meals/semester meal plan (including weekends: OR = 0.6; 95% CI = 0.5, 0.8; excluding weekends: OR = 0.6; 95% CI = 0.4, 0.8; Table 5).

When females and males were examined together, no association between depressed mood, anxiety, perception that meal plan provided adequate food, parents sending food, roommate sharing food, or friends sharing food, was associated with the number of meals students on the 180 meals/semester meal plan used. However, students on the 180 meals/semester meal plan who reported working a job used fewer meals per week on average in the 28 days prior to survey completion when weekends were included in the models (ß = −1.3, 95% CI = −2.3, −0.3; Table 6) and when weekends were excluded (ß = −0.9, 95% CI = −1.7, −0.2).

When stratifying by gender, females on the 180 meals/semester meal plan who worked a job used fewer meals (β = −1.3, 95% CI = −2.5, −0.2; β = −0.9, 95% CI = −1.7, −0.1; Table 7); this association was not examined among males on the 180 meals/semester meal plan due to low counts. 

## 4. Discussion

This study examined the association between meal plan use and food insecurity over time. We found that food insecurity was associated with the number of meals a students’ meal plan provided, where students were the least likely to report food insecurity if they were on the most expensive (unlimited) meal plan. However, of the students on an unlimited meal plan in the Fall semester, 26% still reported food insecurity at the end of the semester. Surprisingly, students reported food insecurity while having meal plan meals available. There were indications that students who used their meal plan more frequently were less likely to be food insecure. This study indicates that students who work a job have lower meal plan use. From this study, it is unclear why students with unused meal plans, and students on unlimited meal plans in particular, reported food insecurity. 

The extent to which food insecurity was reported by students on an unlimited meal plan, and the finding that the students in this study reporting food insecurity typically had meals available to them via their meal plan, was unexpected. Of concern is the possibility that the USDA food security modules (the US Household Food Security Survey Module, US Adult Food Security Survey Module, and Six-Item Short Form of the Food Security Survey Module) are inappropriate for college student populations. The USDA food security modules are the standard tools used to determine food security in the US. A systematic review of food insecurity among college students that included studies from Australia, Canada, Malaysia, South Africa, and the US reports that 14 of the 17 (82%) studies included in the review used one of the USDA food security modules, or a slight variation of one of the USDA food security modules [2]. If the USDA food security modules do not accurately assess students’ food security status, analyses using the USDA food security modules to examine food insecurity among college students may be inaccurate. While the USDA food security modules have been validated for US households [41], research is needed to ensure the construct and external validity (in particular) of any tool used to examine food insecurity among college students. While this study raises concerns about what the USDA food security modules are measuring for US college students living on campus with a prepaid meal plan, it is also unclear if the USDA food security modules are working as intended for other college populations. Validation studies should be conducted for college student demographics both within and outside the US, as this USDA food security modules have been used in international research among university students [12,42,43,44,45,46,47]. In particular, validation studies should examine students at universities which (a) require students to purchase a meal plan, (b) provide students with the option of purchasing a meal plan, and (c) do not provide meal plans. 

The USDA food security module used in this study may have misclassified some students as the questions asked were not appropriate for this demographic. Several of the food insecurity screener questions refer to not having enough money for food, and students may not think of their prepaid meals as a ‘money’ source. Other questions in the food insecurity screener may not be applicable to students with a meal plan. For instance, cutting the size of a meal and eating less may not be applicable when eating at an all-you-can-eat dining hall. Potentially, questions such as “In the past one month, could you have eaten balanced meals every day (even if it meant changing your typical daily routine, such as getting up earlier)?” and “In the past one month, were you ever hungry but didn’t eat because it wasn’t possible to obtain food?” may be more appropriate for college populations. While some students may have misreported their food security status as lower than it actually was (potentially due to misinterpretation of the questions), another study has suggested that the USDA 18-item food security module may under-report, not over-report, food insecurity [48]. It is unclear if the USDA 6-item food security module used in this study also under-reports food insecurity. Even if the USDA food security module is misclassifying some students, the systematic differences observed for college students’ health and academic outcomes by food security status [12,14,16] indicate that, in general, students who report food insecurity struggle more than students who report food security. Future research should estimate what proportion of false positives and false negatives occurs among student populations when the USDA food security modules are used. 

Students may report food insecurity while having meals available to them for many reasons. Potentially, some of the students who reported food insecurity *perceived* less access to food than their food secure counterparts. Food insecure students typically report higher levels of depression [17,18], and one study found that students who report depression also report more *perceived* barriers to healthful eating [49]. However, there was no suggestion that depression was associated with lower meal plan use in this current study. Additionally, students reporting food insecurity may have eating patterns that do not fit well with the dining hall hours. Prior studies have shown that, in general, students reporting food insecurity obtain fewer hours of sleep [19] and report higher levels of stress [13,17]—which has been associated with night eating [50]. While food insecure and secure students used their meal plan to a similar extent, some students may have reported food insecurity due to difficulty in obtaining additional food at night (when most of the university food outlets are closed). If students are truly food insecure, and have meals available to them, understanding why these apparently contradictory findings co-occur could help to greatly reduce the prevalence of food insecurity being reported on college campuses. For instance, if students report food insecurity while having meals available to them due to time management and transportation issues, providing assistance in these areas could help some of these students to become food secure. Future research should closely examine students’ self-efficacy for food security and include discussions with students to determine the situation(s) that led to food insecurity. Future research should also examine how food insecurity is associated with the meals other types of university students (e.g., older students, commuting students, and/or international contexts) consume on campus. 

This study suggests that at the end of the semester, students with higher meal plan use are less likely to report food insecurity. Many of the students used, on average, one meal from their meal plan per day. It is unclear how many non-dining hall meals the students consumed in addition to this one meal, and if students had food options available to them when they wanted them. If a student typically has one large dining hall meal each day, failure to obtain this meal for just one day a week may result in the student reporting food insecurity. However, we note that at this university, there are several dining halls, each open for around 12 h on weekdays (and around 8 h on weekends). In addition to the dining hall locations, all students are able to use one meal exchange per day at select locations on campus: A location offering a meal exchange was open until 2AM each day. As such, this university provides students with many dining options; however, at evening/night, some students would have to leave their residence hall and walk 5–10 min to use their meal plan. It was unclear how busy the locations that provided meals late at night were, and how long the wait time to obtain a meal at these locations typically was. If students are struggling to use their meal plan, this finding may indicate that students’ time management, rather than university dining options, is a key issue with student food insecurity. The typical frequency and duration of students’ food insecurity should be examined; for instance, future studies should determine if students’ food insecurity tends to be chronic or acute, and how students’ experiences with food insecurity compare to adults’ experiences with food insecurity. Students’ schedules should also be examined, to determine if healthful meal options at convenient locations are available during the times students are wanting to consume meals.

Students on the 180 meals/semester meal plan who reported food insecurity used significantly fewer meals; however, when examining males and females separately, this effect was only found among males. Potential reasons for lower meal plan use for students on the 180 meals/semester meal plan were examined. Students on the 180 meals/semester meal plan who reported working a job had lower meal plan use. Given the low number of males who reported working a job, no analysis for the male-specific association for working a job and meal plan use was conducted. For the female-specific analysis, working a job was associated with lower meal plan use. 

While the location, days, and hours that students worked was unclear from this study, this suggests that students who work are unable to use their meal plan to the same extent as students who do not work. Broton et al. report that students who are employed are significantly more likely than their non-working counterparts to state that they do not having enough time to eat [22]. As well as having limited time to eat, students are potentially working off campus, and as such unable to use their meal plan during their working hours. Students may also finish work after the main dining halls close, making it more difficult for these students to use their meal plan. Future studies should further examine the association between meal plan use, working a job, and gender. In particular, future studies should examine if students’ meal plan use is associated with the shifts students work. 

Female students who worked used around 1.3 fewer meals per week, which equates to roughly 0.93 fewer meals during the weekdays if the effect is constant across all days. The model suggests a constant effect as students who work were predicted to use 0.9 fewer meals over the weekdays. Peer-reviewed studies have reported that going home for weekends is common among first-year Hispanic [51] and first-generation Appalachian students [52]. A New York Times article also reported many students attending a North-Eastern college leave campus over the weekend [53]. While this study was based in the US Southwest, potentially many students also left the campus on the weekend. From this study, it was unclear if students who worked were off-campus and unable to use their meal plan, or on-campus but choosing to dine elsewhere, and if the reason students’ who worked used fewer meals was the same for weekdays and weekends. 

### 4.1. Strengths/Limitations

This study examines food insecurity from a new angle and presents some surprising results. Strengths of this study include the objective information on meal plan purchase and use, and the large sample of first-year students. Another strength is the analysis of four different meal plans, which shows the odds of food insecurity increasing as the number of meals provided decreases. While this study is not a validation study, the association between lower meal plan use and food insecurity found in this study provides partial support for convergent validity of food insecurity among college students, which is a strength of the study. While it is a strength of this dissertation that the USDA 6-item food security module was used to determine students’ food security status, given our concern around what the USDA food security module is capturing, we also count the use of this module as a limitation. Other limitations of this study are that the findings are from a specific US university, only for first-year students, and during only one academic year (2015–2016). The findings may be different at other universities, in other countries, for students past their first year in college or with different demographics, and for other academic years. While students’ food security status was determined using the validated USDA short form, food security status was determined via only one mode (questionnaire) and only asked once at the end of each semester. Low food secure and very low food secure students were grouped in the present analysis; differences may be more apparent when comparing students with very low food security to food secure students. The number of credits students were enrolled in, students’ GPA, and students’ class schedules, were not examined. Potentially, students also had lower meal plan use if they had a heavier course load, spent more time studying, or had classes during meal times. Students were asked about parents/guardians purchasing and sending food to them but were not asked about how often they ate meals at their parents’/guardians’ (or relatives’) home, and some students may have had lower meal plan use due to meals consumed with family members. While objective measures of meal plan use were used, it was not possible to verify the meal plan information from a secondary source. However, even if many of the meals students used went unreported, the majority of students would still have had meals available to them.

### 4.2. Implications

Students are reporting food insecurity while having prepaid meals available to them. Better understanding why students are not fully using their meal plan could help to develop interventions to reduce the prevalence of food insecurity among college students. Qualitative student interviews may provide necessary insight as to why these two apparent contradictory findings—food insecurity and unused meals—are occurring. Research is needed to understand how to encourage or enable students to use their meal plan more; doing so could reduce the extent of food insecurity on college campuses. 

## Figures and Tables

**Table 1 nutrients-11-00904-t001:** Demographics and key variables by the meal plan purchased in the Fall 2015 (*n* = 470), and Spring 2016 (*n* = 324) semesters.

Variable	Fall 2015					Spring 2016				
	8 meals/week	180 meals/semester ^G^	14 meals/week	Unlimited Meals	*p*. Value	8 meals/week	180 meals/semester ^G^	14 meals/week	Unlimited Meals	*p*. Value
***N***	138	95	155	82		114	84	91	35	
**Food security status at start of semester, *n* (%)**					0.073					0.124
Not FI ^A^	97 (70.3)	70 (73.7)	111 (71.6)	70 (85.4)		69 (60.5)	60 (71.4)	63 (69.2)	28 (80.0)	
FI	41 (29.7)	25 (26.3)	44 (28.4)	12 (14.6)		45 (39.5)	24 (28.6)	28 (30.8)	7 (20.0)	
**Food security status at end of semester, *n* (%)**					0.135					**0.001**
Not FI	81 (58.7)	60 (63.2)	99 (63.9)	61 (74.4)		61 (53.5)	58 (69.0)	61 (67.0)	31 (88.6)	
FI	57 (41.3)	35 (36.8)	56 (36.1)	21 (25.6)		53 (46.5)	26 (31.0)	30 (33.0)	4 (11.4)	
**Age (mean (sd))**	18.3 (0.4)	18.4 (0.4)	18.4 (0.4)	18.5 (0.6)	0.094	18.3 (0.4)	18.4 (0.5)	18.3 (0.4)	18.5 (0.5)	0.270
**Sex, *n* (%)**					**<0.001**					**<0.001**
Female	108 (78.3)	73 (76.8)	111 (71.6)	42 (51.2)		93 (81.6)	63 (75.0)	65 (71.4)	10 (28.6)	
Male	30 (21.7)	22 (23.2)	44 (28.4)	40 (48.8)		21 (18.4)	21 (25.0)	26 (28.6)	25 (71.4)	
**Race/ethnicity, *n* (%)**					0.352					0.695
White	64 (46.4)	52 (54.7)	69 (44.5)	42 (51.2)		45 (39.5)	33 (39.3)	41 (45.1)	18 (51.4)	
Black	13 (9.4)	7 (7.4)	21 (13.5)	7 (8.5)		11 (9.6)	10 (11.9)	14 (15.4)	4 (11.4)	
Hispanic	36 (26.1)	21 (22.1)	37 (23.9)	12 (14.6)		37 (32.5)	23 (27.4)	20 (22.0)	6 (17.1)	
Other	25 (18.1)	15 (15.8)	28 (18.1)	21 (25.6)		21 (18.4)	18 (21.4)	16 (17.6)	7 (20.0)	
**Pell Grant recipient, *n* (%)**					**<0.001**					**0.019**
No	65 (47.1)	57 (60.0)	115 (74.2)	56 (68.3)		56 (49.1)	50 (59.5)	62 (68.1)	25 (71.4)	
Yes	73 (52.9)	38 (40.0)	40 (25.8)	26 (31.7)		58 (50.9)	34 (40.5)	29 (31.9)	10 (28.6)	
**First-generation college student, *n* (%)**					**<0.001**					**0.001**
No	54 (39.1)	60 (63.2)	105 (67.7)	51 (62.2)		46 (40.4)	54 (64.3)	59 (64.8)	21 (60.0)	
Yes	84 (60.9)	35 (36.8)	50 (32.3)	31 (37.8)		68 (59.6)	30 (35.7)	32 (35.2)	14 (40.0)	
**Campus, *n* (%)**					0.815					0.342
A	67 (48.6)	43 (45.3)	67 (43.2)	39 (47.6)		47 (41.2)	27 (32.1)	34 (37.4)	17 (48.6)	
B	71 (51.4)	52 (54.7)	88 (56.8)	43 (52.4)		67 (58.8)	57 (67.9)	57 (62.6)	18 (51.4)	
**Depressed mood, *n* (%)**					0.748					0.379
Low	56 (40.6)	39 (41.1)	68 (43.9)	39 (47.6)		44 (38.6)	38 (45.2)	37 (40.7)	19 (54.3)	
High	82 (59.4)	56 (58.9)	87 (56.1)	43 (52.4)		70 (61.4)	46 (54.8)	54 (59.3)	16 (45.7)	
**Anxiety, *n* (%) ^B^**					0.700					0.323
No	125 (90.6)	83 (87.4)	133 (86.4)	71 (86.6)		99 (86.8)	75 (89.3)	73 (80.2)	31 (88.6)	
Yes	13 (9.4)	12 (12.6)	21 (13.6)	11 (13.4)		15 (13.2)	9 (10.7)	18 (19.8)	4 (11.4)	
**Works a job, *n* (%) ^C^**					**0.041**					0.645
No	92 (66.7)	70 (75.3)	124 (80.0)	64 (80.0)		62 (54.9)	46 (54.8)	54 (59.3)	23 (65.7)	
Yes	46 (33.3)	23 (24.7)	31 (20.0)	16 (20.0)		51 (45.1)	38 (45.2)	37 (40.7)	12 (34.3)	
**Meal plan provides adequate food, *n* (%)**					**0.002**					0.679
Strongly disagree/disagree	46 (33.3)	18 (18.9)	29 (18.7)	11 (13.4)		30 (26.3)	18 (21.4)	22 (24.2)	6 (17.1)	
Agree/strongly agree	92 (66.7)	77 (81.1)	126 (81.3)	71 (86.6)		84 (73.7)	66 (78.6)	69 (75.8)	29 (82.9)	
**Parents send food, *n* (%) ^D^**					**<0.001**					**<0.001**
Less than twice a semester	38 (27.7)	45 (47.9)	68 (44.4)	46 (56.1)		39 (34.2)	42 (50.0)	56 (61.5)	23 (65.7)	
Twice a semester or more	99 (72.3)	49 (52.1)	85 (55.6)	36 (43.9)		75 (65.8)	42 (50.0)	35 (38.5)	12 (34.3)	
**Roommate shares food with participant, *n* (%) ^E^**					0.990					0.154
Never/rarely	69 (51.1)	48 (51.1)	80 (52.6)	41 (50.6)		58 (51.3)	50 (60.2)	56 (61.5)	25 (71.4)	
Sometimes/often	66 (48.9)	46 (48.9)	72 (47.4)	40 (49.4)		55 (48.7)	33 (39.8)	35 (38.5)	10 (28.6)	
**Friends share food with participant, *n* (%) ^F^**					0.819					0.865
Never/rarely	47 (34.8)	29 (30.9)	49 (32.5)	30 (37.0)		48 (42.5)	33 (39.8)	40 (44.9)	16 (47.1)	
Sometimes/often	88 (65.2)	65 (69.1)	102 (67.5)	51 (63.0)		65 (57.5)	50 (60.2)	49 (55.1)	18 (52.9)	

^A^ FI = Food insecure; ^B^
*n* = 469 Fall, 324 Spring; ^C^
*n* = 466 Fall, 323 Spring; ^D^
*n* = 466 Fall, 324 Spring; ^E^
*n* = 462 Fall, 322 Spring; ^F^
*n* = 461 Fall, 319 Spring; ^G^ 180 meals/semester equates to approximately 11 meals per week with consistent use. Bold indicates statistical significance (*p* < 0.05).

**Table 2 nutrients-11-00904-t002:** The number of meals students used from their meal plan in the 28 days prior to survey, and across the semester (*n* = 534; mean (sd)).

Meal Plan Use	Fall 2015				Spring 2016			
	8 meals/week	180 meals/semester ^A^	14 meals/week	Unlimited Meals	8 meals/week	180 meals/semester ^A^	14 meals/week	Unlimited Meals
**Females**								
***N***	108	73	111	42	93	63	65	10
**Number of meals in 28 days prior to survey**								
Including guest passes	22.6 (9.3)	30.5 (14.5)	33.4 (12.1)	40.4 (18.9)	20.4 (8.4)	30.6 (12.0)	29.5 (11.4)	41.3 (23.7)
Excluding guest passes	20.1 (7.8)	30.4 (14.5) ^B^	30.9 (10.0)	38.1 (17.3)	19.0 (7.2)	30.6 (12.0) ^B^	27.8 (10.4)	36.1 (21.3)
Excluding guest passes and weekends	16.0 (6.4)	22.0 (9.9)	23.1 (7.5)	28.4 (12.1)	16.0 (5.9)	24.8 (9.2)	22.1 (8.0)	29.7 (16.7)
**Number meals per semester**								
Including guest passes	87.0 (31.0)	111.1 (43.2)	130.5 (42.5)	164.0 (58.6)	86.2 (31.0)	125.1 (41.9)	123.6 (39.2)	157.0 (77.6)
Excluding guest passes	79.0 (26.3)	111.0 (43.2)	123.3 (38.1)	155.9 (54.6)	78.7 (26.6)	125.0 (41.9)	117.2 (35.5)	141.0 (70.6)
Excluding guest passes and weekends	65.5 (20.7)	87.5 (32.5)	97.1 (28.4)	123.4 (40.5)	66.7 (22.4)	103.7 (33.4)	95.9 (26.6)	118.6 (57.8)
**Percentage of meals used ^C^**								
28 days prior to survey	71%	68%	60%		64%	73%	53%	
Per semester ^D^	68%	62%	58%		63%	70%	52%	
**Males**								
***N***	30	22	44	40	21	21	26	25
**Number of meals in 28 days prior to survey**								
Including guest passes	24.9 (7.2)	33.3 (13.0)	41.6 (13.3)	59.5 (23.9)	24.0 (7.3)	35.0 (11.6)	37.3 (11.8)	64.0 (25.9)
Excluding guest passes	23.0 (6.1)	33.3 (13.0)	36.6 (9.7)	57.2 (22.2)	22.0 (5.4)	35.0 (11.6)	34.2 (10.2)	61.6 (23.2)
Excluding guest passes and weekends	18.4 (4.7)	24.9 (9.5)	26.6 (7.6)	42.6 (15.7)	17.9 (4.8)	28.0 (9.5)	25.8 (6.9)	47.8 (17.4)
**Number meals per semester**								
Including guest passes	99.3 (30.3)	122.5 (45.6)	158.1 (42.1)	232.0 (86.2)	105.7 (22.5)	132.8 (40.9)	154.5 (48.2)	253.7 (94.9)
Excluding guest passes	89.4 (26.3)	122.5 (45.6)	146.4 (35.9)	224.3 (82.9)	94.0 (19.2)	132.4 (40.7)	142.6 (41.8)	246.0 (88.2)
Excluding guest passes and weekends	73.9 (20.4)	99.0 (36.0)	111.3 (27.5)	178.0 (62.6)	80.2 (14.3)	108.8 (34.3)	113.8 (30.5)	198.9 (70.2)
**Percentage of meals used ^C^**								
28 days prior to survey	78%	74%	74%		75%	83%	67%	
Per semester ^D^	78%	68%	71%		78%	74%	65%	

The number of meals males and females used in each semester was significantly different by meal plan (*p* < 0.001); ^A^ 180 meals/semester equates to approximately 11 meals per week with consistent use; ^B^ The 180 meals/semester meal plan does not allow guest meals. A few students (*n* = 4 Fall, *n* = 7 Spring) used a guest pass prior to changing onto the 180 meals/semester meal plan. A few students on the 180 meals/semester meal plan swiped their meal plan twice in the same minute; the second swipe was classified as a guest meal. ^C^ The percentage of meals used was based on the number of meals used (including guest passes) out of the theoretical maximum number of meals a student on each meal plan could use. For students on an 8 (14) meals/week meal plan, the theoretical maximum number of meals a student could have used was: 32 (56) meals in 28 days, 128 (224) meals in the Fall semester, and 136 (238) meals in the Spring semester. The percentage of meals used in the 28 days prior to the survey for students on the 180 meals/semester meal plan was based off a theoretical maximum of 45 meals for the Fall semester and 42 meals for the Spring semester. ^D^ The semester length was calculated as 16 weeks for the Fall semester and 17 weeks for the Spring semester. This is less than the actual semester lengths (18 weeks) to account for less purchases made in the first and last weeks of semester, and four days of missing data during the Fall semester.

**Table 3 nutrients-11-00904-t003:** Results of the logistic generalized estimating equation (GEE) examining food insecurity at the end of the semester by sociodemographics and meal plan (*n* = 534 participants, 794 data points).

Variable	OR	95% CI	*p*. Value
**Meal plan**			
Unlimited	(ref)		
8 meals/week	2.2	(1.2, 4.1)	**0.016**
180 meals/semester ^A^	1.6	(0.8, 3.0)	0.176
14 meals/week	1.4	(0.8, 2.7)	0.247
**Food security at start of semester**			
Food secure	(ref)		
Food insecure	8.8	(6.0, 12.8)	**<0.001**
**Sex**			
Female	(ref)		
Male	0.9	(0.6, 1.3)	0.472
**Race/ethnicity**			
White	(ref)		
Black	1.7	(0.9, 3.1)	0.085
Hispanic	1.0	(0.6, 1.7)	0.881
Other race/ethnicity	0.6	(0.4, 1.0)	0.052
**Pell Grant recipient**	1.0	(0.6, 1.5)	0.932
**First-generation college student**	0.8	(0.6, 1.2)	0.373
**Campus**			
A	(ref)		
B	0.7	(0.5, 1.0)	**0.049**
**Spring 2016 semester**	0.6	(0.4, 0.9)	**0.012**
**Week of semester**	1.2	(1.0, 1.4)	**0.026**

^A^ 180 meals/semester equates to approximately 11 meals per week with consistent use. Bold indicates statistical significance (*p* < 0.05).

**Table 4 nutrients-11-00904-t004:** Results of the logistic generalized estimating equation (GEE) determining if food insecurity at the end of the semester is associated with number of meals used ^A^.

Meal Plan	Number of Students	Number of Data Points	Food Insecure at End of the Semester					
			Including Weekends			Excluding Weekends		
			OR	95% CI	*p*. Value	OR	95% CI	*p*. Value
8 meals/week	175	252	0.9	(0.8, 1.1)	0.476	0.9	(0.7, 1.1)	0.343
180 meals/semester ^B^	145	179	0.9	(0.8, 1.0)	**0.011**	0.8	(0.7, 0.9)	**0.004**
14 meals/week	184	246	0.9	(0.8, 1.0)	0.150	0.9	(0.7, 1.0)	0.081
Unlimited meals	86	117	1.0	(0.9, 1.1)	0.895	1.0	(0.8, 1.1)	0.612

Bold indicates statistical significance (*p* < 0.05). A control for food security status at start of semester, sex, race/ethnicity, Pell Grant status, first-generation status, campus, semester, week survey was completed, and a clustering effect by student were included in the models. ^A^ Number of meals used is determined by the number of meals used in the 28 days prior to survey completion at the end of the semester and divided by four to obtain an average number of meals per week at the end of the semester. ^B^ 180 meals/semester equates to approximately 11 meals per week with consistent use.

**Table 5 nutrients-11-00904-t005:** Results of the logistic generalized estimating equation (GEE) determining if food insecurity at the end of the semester is associated with number of meals used ^A^ after stratifying by gender and meal plan.

Meal Plan	Number of Students	Number of Data Points	Food Insecure at End of the Semester					
			Including Weekends			Excluding Weekends		
			OR	95% CI	*p*. Value	OR	95% CI	*p*. Value
**Females**								
8 meals/week	142	201	0.9	(0.8, 1.1)	0.414	0.9	(0.7, 1.1)	0.456
180 meals/semester ^B^	109	136	0.9	(0.8, 1.1)	0.253	0.9	(0.7, 1.0)	0.136
14 meals/week	133	176	0.9	(0.8, 1.1)	0.467	0.9	(0.7, 1.1)	0.316
Unlimited meals	43	52	1.0	(0.9, 1.2)	0.763	1.0	(0.8, 1.3)	0.943
**Males**								
8 meals/week	33	51	1.1	(0.7, 1.6)	0.766	0.9	(0.6, 1.5)	0.774
180 meals/semester ^B^	36	43	0.6	(0.5, 0.8)	**0.002**	0.6	(0.4, 0.8)	**0.002**
14 meals/week	51	70	0.8	(0.5, 1.1)	0.097	0.7	(0.4, 11)	0.084
Unlimited meals	43	65	0.9	(0.8, 1.1)	0.381	0.9	(0.7, 1.1)	0.221

A control for food security status at start of semester, campus, semester, and a clustering effect by student were included in the models. Due to the small sample sizes, sex, race/ethnicity, Pell Grant status, first-generation status, and week survey was completed were not included in the models. ^A^ Number of meals used is determined by the number of meals used in the 28 days prior to survey completion at the end of the semester and divided by four to obtain an average number of meals per week at the end of the semester. ^B^ 180 meals/semester equates to approximately 11 meals per week with consistent use. Bold indicates statistical significance (*p* < 0.05).

**Table 6 nutrients-11-00904-t006:** Results of the linear generalized estimating equation (GEE) testing what factors are related to the number of meals students used ^A^ on the 180 meals/semester ^B^ meal plan.

Variable	Number of Students	Number of Data Points	Number of Meals Used ^A^					
			Including Weekends			Excluding Weekends		
			B	95% CI	*p*. Value	β	95% CI	*p*. Value
Depressed mood	*n* = 145	179	−0.5	(−1.4, 0.4)	0.295	−0.2	(−0.9, 0.4)	0.533
Anxiety	*n* = 145	179	0.4	(−1.1, 1.9)	0.622	0.3	(−0.9, 1.4)	0.658
Works a job	*n* = 143	177	−1.3	(−2.3, −0.3)	**0.012**	−0.9	(−1.7, −0.2)	**0.013**
Meal plan provides adequate food	*n* = 145	179	0.0	(−1.2, 1.1)	0.988	0.0	(−0.8, 0.9)	0.984
Parents send food	*n* = 144	178	0.2	(−0.8, 1.1)	0.734	0.2	(−0.5, 0.9)	0.593
Roommate shares food with participant	*n* = 143	177	0.3	(−0.7, 1.3)	0.599	0.2	(−0.6, 0.9)	0.668
Friends share food with participant	*n* = 143	177	0.5	(−0.5, 1.4)	0.320	0.5	(−0.2, 1.2)	0.200

A control for sex, race/ethnicity, Pell Grant status, first-generation status, campus, semester, and a clustering effect by student were included in the models. ^A^ Number of meals used is determined by the number of meals used in the 28 days prior to survey completion at the end of the semester and divided by four to obtain an average number of meals per week at the end of the semester. ^B^ 180 meals/semester equates to approximately 11 meals per week with consistent use. Bold indicates statistical significance (*p* < 0.05).

**Table 7 nutrients-11-00904-t007:** Results of the linear generalized estimating equation (GEE) testing what factors are related to the number of meals students used ^A^ on the 180 meals/semester ^B^ meal plan after stratifying by gender.

Variable	Number of Students	Number of Data Points	Number of Meals Used ^A^					
			Including Weekends			Excluding Weekends		
			B	95% CI	*p*. Value	β	95% CI	*p*. Value
**Female**								
Depressed mood	109	136	-0.5	(−1.6, 0.5)	0.316	−0.3	(−1.0, 0.5)	0.516
Anxiety	109	136	0.2	(−1.4, 1.7)	0.829	0.1	(−1.0, 1.2)	0.836
Works a job	108	135	−1.3	(−2.5, −0.2)	**0.019**	−0.9	(−1.7, −0.1)	**0.034**
Meal plan provides adequate food	109	136	0.6	(−0.8, 1.9)	0.411	0.3	(−0.7, 1.3)	0.539
Parents send food	108	135	0.3	(−0.8, 1.4)	0.585	0.3	(−0.5, 1.1)	0.433
Roommate shares food with participant	108	135	0.0	(−1.2, 1.2)	0.974	−0.1	(−0.9, 0.8)	0.869
Friends share food with participant	108	135	0.2	(−0.9, 1.3)	0.683	0.3	(−0.5, 1.1)	0.436
**Male**								
Depressed mood	36	43	−0.8	(−2.5, 0.9)	0.348	−0.4	(−1.7, 0.9)	0.561
Anxiety ^C^	36	43						
Works a job ^C^	35	42						
Meal plan provides adequate food ^C^	36	43						
Parents send food	36	43	−0.2	(2.3, 1.9)	0.845	−0.2	(−1.8, 1.5)	0.854
Roommate shares food with participant	35	42	1.2	(−0.5, 3.0)	0.161	0.9	(−0.5, 2.2)	0.223
Friends share food with participant	35	42	0.8	(−0.9, 2.6)	0.358	0.6	(−0.8, 2.0)	0.419

A control for campus, semester, and a clustering effect by student were included in the models. Due to the small sample sizes, sex, race/ethnicity, Pell Grant status, first-generation status, and week survey was completed were not included in the models. ^A^ Number of meals used is determined by the number of meals used in the 28 days prior to survey completion at the end of the semester and divided by four to obtain an average number of meals per week at the end of the semester. ^B^ 180 meals/semester equates to approximately 11 meals per week with consistent use. ^C^ Due to low counts in the number of males reporting low anxiety (*n* = 0 in Fall 2015 and Spring 2016), working a job (*n* = 2 in Fall 2015, *n* = 6 in Spring 2016), and perceiving the meal plan did not provide adequate food (*n* = 3 in Fall 2015, *n* = 5 in Spring 2016), the analyses for anxiety, working a job, and perception that meal plan provides adequate food were not run for males. Bold indicates statistical significance (*p* < 0.05).

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
