# Peer review of "Food Insecure College Students and Objective Measurements of Their Unused Meal Plans"

_nutrients, 2019, doi:10.3390/nu11040904_

Round 1

Reviewer 1 Report

This study elucidated factors asscoiated with colledge student foood insecurity, and provided an new infromation instead of some limits (locality, and the questionnare for insecurity).

For readers of the Nutrients (maybe, all over the world),
 please add the nation name in the Abstract,
            the university name (one university ASU?, or several universities) in the Methods,
         and the start of the school year, and semetester in the Methods
               (Schools of 52% nations start in Semtember, but others do in different months),
  and explain a GPA (academic performance?).

Ref 47. "A New York Times article also reported many students attending a North-Eastern college leave campus over the weekend [47]." Is this university located in a North-East area?

Could the authors provide suggestions for students of colledges that do not have a meal plan, or do not ask them to live on campus? The results are applicable only in US?

Author Response

Reviewer 1:

This study elucidated factors asscoiated with colledge student foood insecurity, and provided an new infromation instead of some limits (locality, and the questionnare for insecurity).

For readers of the Nutrients (maybe, all over the world), 
 please add the nation name in the Abstract,
            the university name (one university ASU?, or several universities) in the Methods,
         and the start of the school year, and semetester in the Methods 
               (Schools of 52% nations start in Semtember, but others do in different months),
  and explain a GPA (academic performance?).
We thank the reviewer for these suggestions and have made the requested changes from above in the document.

Ref 47. "A New York Times article also reported many students attending a North-Eastern college leave campus over the weekend [47]." Is this university located in a North-East area?
We thank the reviewer for this question. The university this data was obtained from was from a South-Western university. We have added a statement “While this study was based in the US southwest, potentially many students also left the campus on the weekend.” on page 16 to clarify why results from a different region were included.

Could the authors provide suggestions for students of colledges that do not have a meal plan, or do not ask them to live on campus? The results are applicable only in US?
We appreciate this comment as both reviewers indicated an interest on the implications of the findings among non-US educational systems. In the discussion we have discussed that the USDA food security tool has not been validated for college students – and that validation for students with a meal plan, and other students, should be conducted.

On page 14 we now state “While the USDA food security modules have been validated for US households [41], research is needed to ensure the construct and external validity (in particular) of any tool used to examine food insecurity among college students. While this study raises concerns about what the USDA food security modules are measuring for US college student living on campus with a pre-paid meal plan, it is also unclear if the USDA food security modules are working as intended for other college populations.  Validation studies should be conducted for college student demographics both within and outside the US, as this USDA food security modules have been used in international research among university students [12, 42-47]. In particular, validation studies should examine students at universities which a) require students to purchase a meal plan, b) provide students the option of purchasing a meal plan, and c) do not provide meal plans.”

On page 15 we have also added a discussion about food availability when students’ are wishing to consume meals. If healthful meal options are not open when students’ are looking to consume meals, this is a problem. This is relevant to both US, and non-US, students. “It is unclear how many non-dining hall meals the students consumed in addition to this one meal, and if students’ had food options available to them when they wanted them”,  “Students’ schedules should also be examined, to determine if healthful meal options at convenient locations are available during the times students are wanting to consume meals.”

On page 15 we also mention that the meals other types of university students (e.g., not first-year US college students living on campus with a meal plan) consume while on campus should be examined, with differences by food security status examined. We state. “Future research should also examine how food insecurity is associated with the meals other types of university students (e.g., older students, commuting students, and/or international contexts) consume on campus.”

We have also added to the limitations (page 16) that this study was only US-based. “Other limitations of this study are that the findings are from a specific US university, only for first-year students, and during only one academic year (2015-2016). The findings may be different at other universities, in other countries, for students past their first year in college or with different demographics, and for other academic years.”

Reviewer 2 Report

It is an exhaustive study based on data referred by students in a survey. It is to praise the intensity with which he analyzes them.

In his study he does not offer any objective or subjective result related to food insecurity that he diagnoses simply by a survey. It does not relate its results with nutritional data such as BMI or anemia or even iron deficiency. Nor does it relate to academic results. We can not say that the data presented has a clinical effect. The high prevalence of food insecurity among college students is of concern because food insecurity is associated with worse health outcomes [9-11], lower academic performance [12-16], higher stress [13, 17], higher depression [13, 17-19], less sleep [19], and a lower GPA [12-16].

For you, "it is unclear why students with unused meal plans, and students on unlimited meal plans in particular, reported food insecurity". I think that in a country like USA, food insecurity does not depend on the accessibility of food, but on the availability of time to take that food or go to the place where it is administered. You are studying the behavior of young people who are more interested in their time than in eating. For them, appearance or even thinness is more important than good nutrition. I think they would improve the results more with information campaigns that facilitate more food.

Another aspect that needs to be addressed is sex: women tend to be on a diet and skip meals to be thin.

Regarding the responses to a food survey we must remember that thin people tend to exaggerate their intake while the obese often forget that they have taken calorie-rich foods out of meals.

I am very surprised by your statement of "If the USDA food security tool does not accurately assess students' food security status, analyze using the USDA food security tool to examine food insecurity among college students may be inaccurate". If you are not sure of the tool you have used, all the results obtained and the important analysis works you have done are just fireworks.

The main limitation of this study is that the results are limited exclusively to the US educational system and I think it has little interest for other countries.

Author Response

Reviewer 2:

It is an exhaustive study based on data referred by students in a survey. It is to praise the intensity with which he analyzes them.
Thankyou!

In his study he does not offer any objective or subjective result related to food insecurity that he diagnoses simply by a survey. It does not relate its results with nutritional data such as BMI or anemia or even iron deficiency. Nor does it relate to academic results. We can not say that the data presented has a clinical effect. The high prevalence of food insecurity among college students is of concern because food insecurity is associated with worse health outcomes [9-11], lower academic performance [12-16], higher stress [13, 17], higher depression [13, 17-19], less sleep [19], and a lower GPA [12-16].
We appreciate this feedback. On page 2 we clarity that food insecurity by itself is of clinical concern. “Food insecurity by itself is of clinical concern [5-8, 9-11, 13, 17-19]. As meal plan use is dietary-related behavior, this study sets the stage for future research to assess differential dietary intake by differential meal plan use.”

For you, "it is unclear why students with unused meal plans, and students on unlimited meal plans in particular, reported food insecurity". I think that in a country like USA, food insecurity does not depend on the accessibility of food, but on the availability of time to take that food or go to the place where it is administered. You are studying the behavior of young people who are more interested in their time than in eating. For them, appearance or even thinness is more important than good nutrition. I think they would improve the results more with information campaigns that facilitate more food.
We thank the reviewer for these comments. In the discussion we have added a comment regarding time and transportation. On page 15 we now state “For instance, if students report food insecurity while having meals available to them due to time management and transportation issues, providing assistance in these areas could help some of these students become food secure.”

Another aspect that needs to be addressed is sex: women tend to be on a diet and skip meals to be thin.
We agree that sex differences are important. We controlled for sex in the models, and then also re-ran the main analyses stratified by sex to further examine any differences by sex (see Table 3, Tables 4 and 5, and Tables 6 and 7)

Regarding the responses to a food survey we must remember that thin people tend to exaggerate their intake while the obese often forget that they have taken calorie-rich foods out of meals.
We thank the reviewer for their thoughts. These issues are beyond the scope of this analysis, as we did not examine dietary intake among obese vs non-obese students.

I am very surprised by your statement of "If the USDA food security tool does not accurately assess students' food security status, analyze using the USDA food security tool to examine food insecurity among college students may be inaccurate". If you are not sure of the tool you have used, all the results obtained and the important analysis works you have done are just fireworks.
We thank the reviewer for this comment. We have clarified that USDA food security assessments are the main tool researchers use to examine food security in the US. The USDA food security assessments, including the one in this study, are routinely used to examine food insecurity among college students, and given our findings, we are concerned about the validity of these questions in assessing food insecurity among university-based populations. On page 14 (directly above the sentence you questioned) we now state “The USDA food security modules are the standard tools used to determine food security in the US. A systematic review of food insecurity among college students that included studies from Australia, Canada, Malaysia, South Africa, and the US, reports that 14 of the 17 (82%) studies included in the review used one of the USDA food security modules, or a slight variation of one of the USDA food security modules [2].”

The main limitation of this study is that the results are limited exclusively to the US educational system and I think it has little interest for other countries.
Thank you for this feedback. We agree that the analysis is US-focused, and have added this as a limitation to the study. College students across the world face food insecurity, and most studies use the USDA food security tools to examine food insecurity among college students. In the discussion, page 14, we now highlight how frequently the USDA food security tools are used globally "“The USDA food security modules are the standard tools used to determine food security in the US. A systematic review of food insecurity among college students that included studies from Australia, Canada, Malaysia, South Africa, and the US, reports that 14 of the 17 (82%) studies included in the review used one of the USDA food security modules, or a slight variation of one of the USDA food security modules [2].”

On page 15 we have also added a discussion about food availability when students’ are wishing to consume meals. If healthful meal options are not open when students’ are looking to consume meals, this is a problem. This is relevant to both US, and non-US, students. “It is unclear how many non-dining hall meals the students consumed in addition to this one meal, and if students’ had food options available to them when they wanted them”,  “Students’ schedules should also be examined, to determine if healthful meal options at convenient locations are available during the times students are wanting to consume meals.”

On page 15 we also mention that the meals other types of university students (e.g., not first-year US college students living on campus with a meal plan) consume while on campus should be examined, with differences by food security status examined. We state. “Future research should also examine how food insecurity is associated with the meals other types of university students (e.g., older students, commuting students, and/or international contexts) consume on campus.”

We now also state in the discussion (page 14) that given the USDA food security tools are not validated for college students, this should be done – both within and outside the US. While this study raises concerns about what the USDA food security modules are measuring for US college student living on campus with a pre-paid meal plan, it is also unclear if the USDA food security modules are working as intended for other college populations.  Validation studies should be conducted for college student demographics both within and outside the US, as this USDA food security modules have been used in international research among university students [12, 42-47]. In particular, validation studies should examine students at universities which a) require students to purchase a meal plan, b) provide students the option of purchasing a meal plan, and c) do not provide meal plans.”
